# In Situ Synthesis of Sn-Beta Zeolite Nanocrystals for Glucose to Hydroxymethylfurfural (HMF)

**Kachaporn Saenluang** [1]**, Anawat Thivasasith** [1,]*[**] , **Pannida Dugkhuntod** [1],
**Peerapol Pornsetmetakul** [1] , **Saros Salakhum** [1]**, Supawadee Namuangruk** [2] **and**
**Chularat Wattanakit** [1]

[1]   Department of Chemical and Biomolecular Engineering, Vidyasirimedhi Institute of Science and Technology,
    School of Energy Science and Engineering, Rayong 21210, Thailand; kachaporn.s_s16@vistec.ac.th (K.S.);
    s15_pannida.d@vistec.ac.th (P.D.); Peerapol.p_s18@vistec.ac.th (P.P.); s15_saros.s@vistec.ac.th (S.S.);
    chularat.w@vistec.ac.th (C.W.)
[2]   National Nanotechnology Center (NANOTEC), National Science and Technology Development Agency,
    Pathum Thai 12120, Thailand; supawadee@nanotec.or.th
[*]   Correspondence: anawat.t@vistec.ac.th; Tel.: +66-3-301-4262

**Abstract:** The Sn substituted Beta nanocrystals have been successfully synthesized by in-situ hydrothermal process with the aid of cyclic diquaternary ammonium (CDM) as the structure-directing agent (SDA). This catalyst exhibits a bifunctional catalytic capability for the conversion of glucose to hydroxymethylfurfural (HMF). The incorporated Sn acting as Lewis acid sites can catalyze the isomerization of glucose to fructose. Subsequently, the Brønsted acid function can convert fructose to HMF via dehydration. The effects of Sn amount, zeolite type, reaction time, reaction temperature, and solvent on the catalytic performances of glucose to HMF, were also investigated in the detail. Interestingly, the conversion of glucose and the HMF yield over 0.4 wt% Sn-Beta zeolite nanocrystals using dioxane/water as a solvent at 120 °C for 24 h are 98.4% and 42.0%, respectively. This example illustrates the benefit of the in-situ synthesized Sn-Beta zeolite nanocrystals in the potential application in the field of biomass conversion.

**Keywords:** in-situ synthesis; Sn-Beta zeolite; isomorphous substitution; glucose; HMF

## 1. Introduction

In recent years, the conversion of biomass to high value-added chemicals, for example, lactic acid, formic acid, levulinic acid, and 5-hydroxymethylfurfural (5-HMF), has been extensively studied [1,2]. In particular, the HMF product has been widely used in the synthesis of many useful compounds, novel polymer materials, plastic resins, and diesel fuel additives [3]. Typically, one of the most promising alternative feedstocks of biomass to produce HMF is glucose, because it is the most abundant monosaccharide and the cheapest hexose, making it as a promising candidate to produce fructose, and subsequently HMF [4]. In a typical procedure, the conversion of glucose to HMF requires the following two main steps: (i) isomerization of glucose to fructose; (ii) the dehydration of fructose to HMF, in which bifunctional catalysts composed of Lewis acid sites and Brønsted acid function are responsible for these two steps, respectively [5,6].

Indeed, there are many types of catalysts that have been used for the conversion of glucose to HMF, such as metal chloride salts (MgCl$_2$, SnCl$_4$) [7], metal oxides (TiO$_2$, ZrO$_2$, Nb$_2$O$_5$) [8,9], metal organic frameworks (ZIF-8) [10], zeolites [11–16], and Sn, Ti and Zr-containing zeolites [17,18]. Among them, the metal incorporated in a zeolite is one of the most important candidates for this reaction because of

high metal dispersion [19], high surface area [20], unique shape selective properties [21], suitable acid properties [22] and high thermal/hydrothermal stability [23], for example.

Various catalysts, such as Sn, Ti and Zr-containing zeolites and especially Sn-incorporated in Beta zeolite framework (Sn-Beta), have been extensively used in the conversion of glucose to fructose [24]. The Sn-Beta zeolite exhibits unique Lewis acidity [17], which is used for the isomerization of glucose to fructose [25]. In addition, by combining with Brønsted acid sites, the dehydration of fructose to HMF has been further proceeded, and therefore, the bifunctional catalysts containing Sn as Lewis acid sites together with Brønsted acid zeolites play an important role in these reaction pathways [26]. Typically, the Sn incorporated Beta zeolite can be produced by following two major strategies: (i) bottom-up approaches, in which the Sn sources are directly added to the zeolite precursor by an in-situ or hydrothermal synthesis process, resulting in Sn being tetrahedrally built into the silica framework of the BEA topology; (ii) top-down approaches, in which the Sn species is deposited on the zeolite surfaces by a post synthesis method via either a wet-impregnation method or an ion-exchange method [27]. However, it seems that the Sn-Beta zeolite obtained via an in-situ or hydrothermal synthesis method has played a very important role because it is a simple method, well-controlled metal dispersion, and is convenient with respect to other methods.

Over the past decade, the Sn-Beta zeolite has been utilized in many catalytic reactions, such as Baeyer-Villiger oxidations [28] and Meerwein-Ponndorf-Verley oxido-reductions [29]. Recently, the Sn-Beta zeolite has been successfully applied in several carbohydrate-related reactions like glucose isomerization to fructose, and subsequently fructose dehydration to HMF [30]. The HMF formation from glucose requires a catalyst containing both Lewis acid function and Brønsted acids site. For example, a physical mixture of Sn Lewis acids and Brønsted acid catalysts can be used as a catalyst [31,32]. It was found that high amount of glucose can be converted to a high yield of HMF in a one-pot reaction system. In addition, Qiang Guo et al. [23] reported that the Amberlyst-131 can act as a Brønsted acid catalyst, and Sn incorporated in Beta zeolite can act as Lewis acid sites. The reaction was carried out in the biphasic phase of dioxane with 5% water content when using fructose or glucose as reactants. It was found that the yield of 5-HMF is up to 74% and 56% from fructose and glucose as reactants, respectively. Moreover, Mark E. Davis et al. [33] also reported that the Sn-Beta zeolite can be used to produce HMF from glucose by a one-pot biphasic water/tetrahydrofuran (THF) reaction system, eventually resulting in achieving the glucose conversions of 79% and HMF selectivity of 72% at 180 °C of the reaction temperature.

As mentioned above, the addition of an organic solvent needs to be considered in the proper selection of HMF extracting solvent (e.g., MIBK, dioxane). Because an organic solvent is an extracting phase in which HMF preferably dissolves with respect to the aqueous solution, the selectivity towards HMF is improved significantly by controlling the production of humins and other by-products during the reaction [34,35]. Simona M et al. [36] also reported that HMF selectivity of 84.3% can be achieved by using Nb-Beta in a biphasic (H$_2$O/tetrahydrofuran (THF)) system. However, without THF as a solvent, the HMF selectivity only reaches 41.7% under an aqueous condition.

In the present study, we reported the facile preparation of Sn incorporated in Brønsted acid Beta nanocrystals obtained via an in-situ hydrothermal synthesis process with the aid of cyclic diquaternary ammonium (CDM) as the structure-directing agent (SDA). By combining the incorporated Sn as Lewis acid sites together with the hierarchical structure of Brønsted acid Beta, it significantly improves the catalytic performance in terms of conversion and selectivity under the aqueous and biphasic phase (dioxane/H$_2$O) for the one-pot synthesis of HMF from glucose. In addition, the effects of Sn amount, zeolite type, reaction time, reaction temperature, and solvent on the conversion catalytic performances of glucose to HMF were also investigated and further discussed in more detail.

## 2. Results and Discussion

### 2.1. Characterization of Synthesized Sn-Beta Nanocrystals

The Sn-incorporated Beta nanocrystals with 0.4 wt% of Sn (0.4 wt% Sn-Beta) have been successfully synthesized by the in-situ hydrothermal process following a modified literature procedure [37,38]. The XRD patterns of all the synthesized samples were used to check the crystalline structure (characteristic) of all catalysts, as shown in Figure 1. Compared with the commercial Beta zeolite (Beta-COM), which was supplied from the Zeolyst International company, the intensities of XRD peaks of the synthesized Sn-Beta nanocrystals are lower than those of the synthesized bare Beta and the commercial Beta zeolite (Beta-COM), indicating that the crystallinity of Beta zeolite nanocrystals was decreased after adding Sn in the zeolite framework [39]. In addition, the XRD pattern has no diffraction peak of Sn due to the small content of Sn (0.4 wt%) or the highly dispersed species inside the Beta zeolite nanocrystal framework [40].

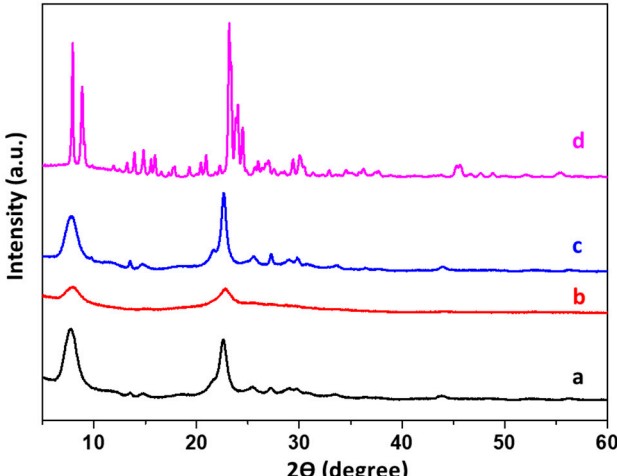

**Figure 1.** XRD patterns of: (**a**) the bare Beta, (**b**) the 0.4 wt% Sn-Beta, (**c**) the commercial Beta (Beta-COM), and (**d**) the conventional ZSM-5 (ZSM-5-CON).

However, the plate-shaped morphology of the 0.4 wt% Sn-Beta sample was not affected by the incorporation of Sn [41] as can be seen in SEM and TEM images (Figure 2A–C,E–G). Obviously, the plate-shaped particle size of the 0.4 wt% Sn-Beta sample and the synthesized bare Beta is approximately 25.3 ± 12 and 21.7 ± 7 nm, respectively. In strong contrast to this, the plate-shaped particle size of the commercial Beta (Beta-COM) is a little bit larger, with the size of 38.3 nm. However, other zeolite frameworks, such as ZSM-5, have also been used for the comparison. In the case of the conventional ZSM-5 sample (ZSM-5-CON), the coffin-shaped crystal is very large, with a particle size of approximately 2500 ± 200 nm. The particle size distribution of all samples is shown in Figure S1. Moreover, to confirm the existence of Sn in the 0.4 wt% Sn-Beta sample, the EDS elemental mapping of Sn was used to observe the Sn component, as shown in Figure 2I.

To further investigate the Sn species of the 0.4 wt% Sn-Beta, the X-ray photoelectron spectroscopy (XPS) was employed as shown in Figure 3. There were three distinct peaks that appeared at 484.4, 486.5, and 487.5 eV, which contributed to $Sn^0$, $SnO_2$, and Sn isomorphous substitution in the zeolite framework, respectively [42,43]. These observations reveal that the 0.4 wt% Sn-Beta obtained by an in-situ hydrothermal process can successfully incorporate Sn active sites into the zeolite framework.

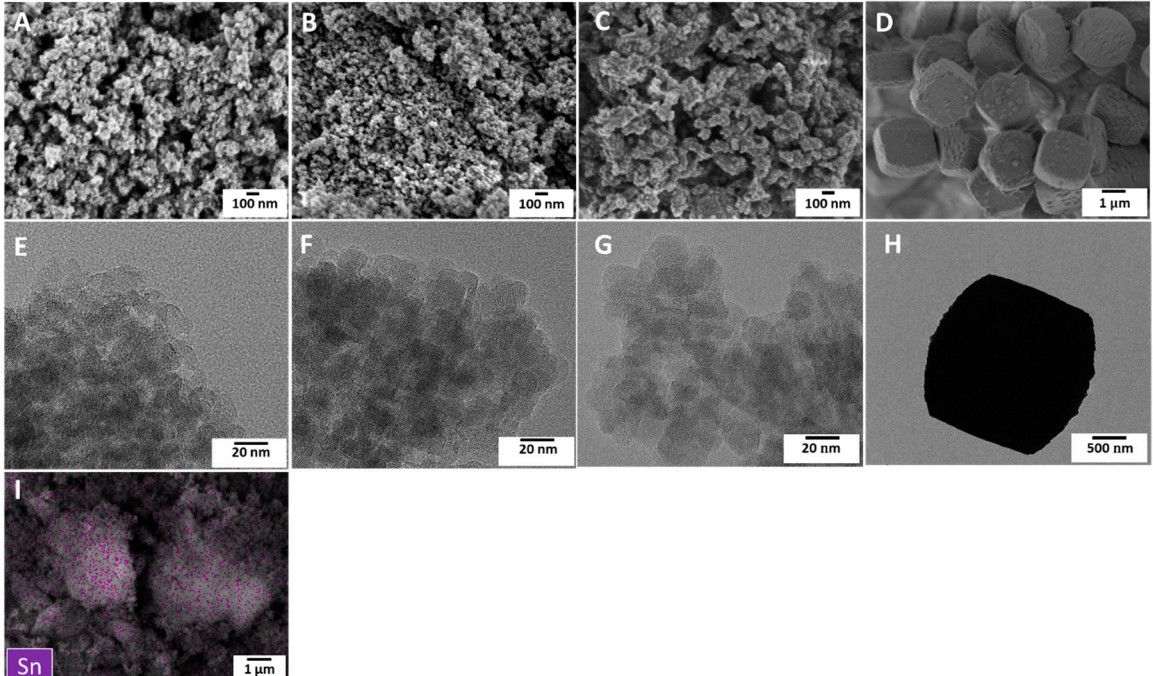

**Figure 2.** SEM and TEM images of: (**A**,**E**) the bare Beta, (**B**,**F**) the 0.4 wt% Sn-Beta, (**C**,**G**) the commercial Beta (Beta-COM), (**D**,**H**) the conventional ZSM-5 (ZSM-5-CON), and (I) the EDS Sn mapping of the in-situ synthesized Sn incorporated Beta (0.4 wt% Sn-Beta).

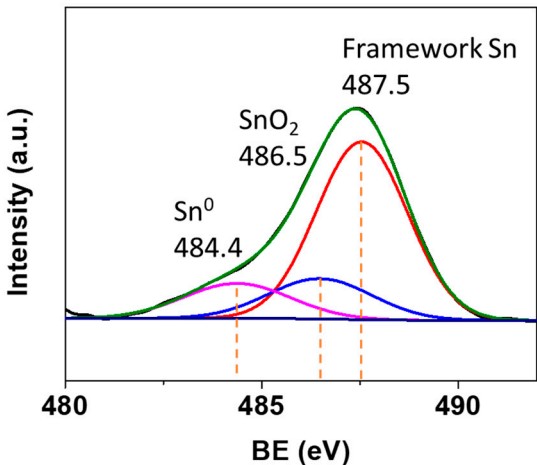

**Figure 3.** XPS spectrum of the 0.4 wt% Sn-Beta representing $Sn^0$, $SnO_2$, and Sn isomorphous substitution in the zeolite framework at 484.4, 486.5, and 487.5 eV, respectively.

To confirm the Si/Al ratio of all samples, the XRF technique was used and it was found that the Si/Al ratios of the bare Beta and 0.4 wt% Sn-Beta are 9.0, and 9.4, respectively. These observations obviously demonstrate that Al sites in the Beta zeolite framework were replaced by Sn, and therefore the Si/Al ratio of the 0.4 wt% Sn-Beta sample is slightly higher than that of the bare Beta [44]. Moreover, the Si/Sn ratio of the 0.4 wt% Sn-Beta sample is approximately 56. Furthermore, the Si/Al ratio of the commercial Beta (Beta-COM) and the conventional ZSM-5 (ZSM-5-CON) was observed as 11.7 and 11.5, respectively as can be seen in Table S1.

To investigate the textural properties of the bare Beta, and the 0.4 wt% Sn-Beta, $N_2$ sorption isotherms and the summarized data are shown in Figure S2 and Table 1, respectively. Obviously, the 0.4 wt% Sn-Beta exhibits a significantly lower BET surface area compared with the bare Beta,

while total pore volume and external pore volume of the 0.4 wt% Sn-Beta is higher than those of the Beta zeolite [41]. These observations also relate to the fact that the low crystallinity of 0.4 wt% Sn-Beta leads to the production of the high external pore volume and the increased total pore volume compared with the synthesized bare Beta. This is similar to what has been described previously [40,45].

**Table 1.** Textural properties of the bare Beta and the 0.4 wt% Sn-Beta.

| Sample | $S_{BET}$ [a] $(m^2/g)$ | $S_{micro}$ [b] $(m^2/g)$ | $S_{ext}$ [c] $(m^2/g)$ | $V_{total}$ [d] $(cm^3/g)$ | $V_{micro}$ [e] $(cm^3/g)$ | $V_{ext}$ [f] $(cm^3/g)$ | $V_{ext}/V_{total}$ [g] |
|---|---|---|---|---|---|---|---|
| Bare Beta | 711 | 670 | 41 | 1.49 | 0.16 | 1.33 | 0.89 |
| 0.4 wt% Sn-Beta | 492 | 470 | 22 | 2.05 | 0.10 | 1.95 | 0.95 |

[a] $S_{BET}$: BET specific surface area. [b] $S_{micro}$: microporous surface area. [c] $S_{ext}$: external surface area. [d] $V_{total}$: total pore volume. [e] $V_{micro}$: micropore volume, [f] $V_{ext} = V_{total} - V_{micro}$; all surface areas and pore volumes are in the units of $m^2 g^{-1}$ and $cm^3 g^{-1}$, respectively. [g] Fraction of external volume.

To evaluate the acid properties, the ammonia temperature-programmed desorption (NH$_3$-TPD) profiles were presented in Figure S3 and the analyzed data are summarized in Table 2. The synthesized bare Beta, the in-situ synthesized Sn incorporated Beta nanocrystals (0.4 wt% Sn-Beta), and the commercial Beta (Beta-COM) demonstrate the similar trend of acid profiles composing of weak (0.237–0.340 mmol g$^{-1}$) and strong acid sites (0.424–0.449 mmol g$^{-1}$) appeared at the temperature in the range of 180, and 300–550 °C, respectively, eventually leading to a similar total acid density in the range of 0.661–0.789 mmol g$^{-1}$. In addition, the conventional ZSM-5 (ZSM-5-CON) demonstrates a slight difference in the densities of weak acid sites, strong acid sites, and total acid sites of 0.440, 0.423 mmol g$^{-1}$, and 0.863 mmol g$^{-1}$, respectively.

Furthermore, to identify the type of acid sites containing Brønsted acid sites (BAS) and Lewis acid sites (LAS) of the 0.4 wt% Sn-Beta and the bare Beta, the adsorption of pyridine monitored by FTIR spectroscopy was performed at 150 °C. It was found that the 0.4 wt%Sn-Beta shows the significant presence of both characteristic signals for Brønsted acid sites ($\nu = 1545$ cm$^{-1}$), and Lewis acid ($\nu = 1455$ cm$^{-1}$) sites. In addition, the band at 1490 cm$^{-1}$ is attributed to pyridine adsorbed on both Brønsted acid sites and Lewis acid sites (BAS+LAS) as shown in Figure 4. Compared with the bare Beta, the B/L ratio of the 0.4 wt% Sn-Beta is significantly decreased, implying that Lewis acid sites increase after Sn insertion into the Beta framework (Table S2). This makes it clear that the 0.4 wt% Sn-Beta composes of both Brønsted acid sites and Lewis acid functions simultaneously and it should be suitable as a catalyst for glucose conversion to HMF, which typically requires the bifunctional catalyst containing Brønsted acid and Lewis acid sites.

**Table 2.** Acid sites density of all samples determined via the ammonia temperature-programmed desorption (NH$_3$-TPD).

| Samples/$T_{max}$ (°C) | Acid Site Density (mmol g$^{-1}$) [a] | | |
|---|---|---|---|
| | Weak (180 °C) | Strong (300–550 °C) | Total |
| Bare Beta | 0.318 | 0.441 | 0.759 |
| 0.4 wt% Sn-Beta | 0.237 | 0.424 | 0.661 |
| Commercial Beta (Beta-COM) | 0.340 | 0.449 | 0.789 |
| Conventional ZSM-5 (ZSM-5-CON) | 0.440 | 0.423 | 0.863 |

[a] The number of acid sites measured by NH3-TPD and analyzed by Gaussian deconvolution.

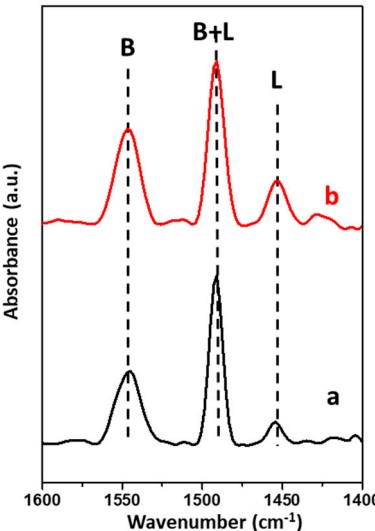

**Figure 4.** FTIR spectra of pyridine adsorbed on: (**a**) the bare Beta, and (**b**) the 0.4 wt% Sn-Beta at 150 °C representing in the adsorption region of Brønsted and Lewis acid sites.

## 2.2. Catalytic Test in the Glucose Conversion to 5-Hydroxymethylfurfural (5-HMF)

To illustrate the beneficial effect of the 0.4 wt% Sn-Beta, the conversion of glucose to 5-HMF in the aqueous phase was initially performed at various reaction times and reaction temperatures, as can be seen in Table 3. Although the addition of Sn into the Beta framework can obviously improve the glucose conversion with respect to the bare Beta, the 0.7 wt% Sn-Beta, does not provide significantly improved catalytic performances with respect to the one with lower Sn loading (0.4 wt%). For example, the glucose conversion and HMF yield over the 0.7 wt% Sn-Beta are 73.8% and 21.5%, respectively, while the glucose conversion and HMF yield over the 0.4 wt% Sn-Beta are 72.0% and 19.1%, respectively.

To further investigate the effect of the reaction time and temperature, the in-situ synthesized Sn-incorporated Beta nanocrystals with 0.4 wt% Sn loading (0.4 wt% Sn-Beta) were used to study in the next step. Obviously, when the reaction time was decreased from 48 h to 24 h, the glucose conversion and HMF yield is decreased from 72.0% to 57.5% and from 19.1% to 15.1%, respectively. As can be seen when decreasing the reaction time from 48 h to 24 h, the HMF yield was decreased only 4.0%, and therefore the 24 h of reaction time is more suitable to leave for a shorter global reaction time. To evaluate the effect of reaction temperature, when increasing the reaction temperature from 120 to 140 °C at 24 h using the 0.4 wt% Sn-Beta catalyst, the glucose conversion was significantly increased from 63.1 to 94.1%. However, the HMF product cannot be produced under this condition because it can be further rehydrated to form formic acid (FA), and levulinic acid (LA) [20,24,31,41].

**Table 3.** Catalytic performances in glucose conversion to 5-hydroxymethylfurfural (5-HMF) in the aqueous phase over the in-situ synthesized Sn-incorporated Beta nanocrystals with various Sn contents obtained at different reaction times and reaction temperatures.

| Samples | Time (h) | T (°C) | Conversion (%) | Product Selectivity (%) | | | | | Yield (%) [g] | MB (%) [h] |
|---|---|---|---|---|---|---|---|---|---|---|
| | | | Glu [a] | Fru [b] | Lac [c] | FA [d] | LA [e] | HMF [f] | | |
| Bare Beta | 48 | 120 | 61.1 | 36.1 | 16.4 | 13.9 | 12.0 | 21.6 | 13.2 | 86 |
| 0.7 wt% Sn-Beta | 48 | 120 | 73.8 | 33.5 | 6.2 | 5.8 | 25.4 | 29.1 | 21.5 | 68 |
| 0.4 wt% Sn-Beta | 48 | 120 | 72.0 | 37.9 | 5.1 | 5.6 | 24.9 | 26.5 | 19.1 | 74 |
| 0.4 wt% Sn-Beta | 24 | 120 | 57.5 | 38.8 | 4.8 | 13.7 | 16.5 | 26.2 | 15.1 | 82 |
| 0.4 wt% Sn-Beta | 24 | 140 | 94.1 | 0.7 | 16.4 | 42.9 | 40 | 0.0 | 0.0 | 67 |

[a] Glucose, [b] Fructose, [c] Lactose, [d] Formic acid, [e] Levulinic acid, [f] Hydroxymethylfurfural, [g] Yield of HMF (%), [h] Mass balance (%).

To further study the effect of solvent on the conversion of glucose to HMF over the 0.4 wt% Sn-Beta, the biphasic phase containing the mixture of dioxane and $H_2O$ was used (Table 4). Interestingly, the catalytic conversion of glucose to HMF over the 0.4 wt% Sn-Beta using a biphasic system at 120 °C and a reaction time of 24 h is significantly improved with the glucose conversion and HMF yield of 98.4% and 42.0%, respectively. Even though the glucose conversion to HMF at a much shorter reaction time is also improved, the glucose conversion and HMF yield at 120 °C for 4 h of the reaction time is approximately 77.3% and 25.0%, respectively. Interestingly, when the glucose conversion to HMF was tested at the lower temperature of 100 °C for 24 h, the glucose conversion and HMF yield were slightly decreased, with the values of 80.3% and 29.0%, respectively, with respect to the one obtained at the reaction temperature of 120 °C for 24 h. However, the glucose conversion to HMF was tested at 100 °C for 4 h and it was found that the glucose conversion and HMF yields are significantly lower with respect to the reaction condition at 100 °C for 24 h. These observations clearly demonstrate that the suitable condition for converting glucose to HMF over the 0.4 wt% Sn-Beta in a biphasic system is the reaction temperature of 100 to 120 °C and a reaction time of 24 h.

**Table 4.** Catalytic performances in glucose conversion to 5-hydroxymethylfurfural (5-HMF) in the biphasic phase (dioxane/water) over different catalysts at different reaction times and reaction temperatures.

| Samples | Phase [a] | Time (h) | T (°C) | Conversion (%) | | Product Selectivity (%) | | | | | Yield (%) [h] | MB (%) [i] |
|---|---|---|---|---|---|---|---|---|---|---|---|---|
| | | | | Glu [b] | Fru [c] | Lac [d] | FA [e] | LA [f] | HMF [g] | | |
| 0.4 wt% Sn-Beta | aq. | 24 | 120 | 57.5 | 38.8 | 4.8 | 13.7 | 16.5 | 26.2 | 15.1 | 82 |
| 0.4 wt% Sn-Beta | bi | 24 | 120 | 98.4 | 5.9 | 42.0 | 8.8 | 0.0 | 42.7 | 42.0 | 61 |
| 0.4 wt% Sn-Beta | bi. | 4 | 120 | 77.3 | 24.3 | 36.3 | 6.6 | 0.5 | 32.3 | 25.0 | 81 |
| 0.4 wt% Sn-Beta | bi. | 24 | 100 | 80.3 | 22.5 | 31.2 | 9.8 | 0.4 | 36.1 | 29.0 | 68 |
| 0.4 wt% Sn-Beta | bi. | 4 | 100 | 30.1 | 75.3 | 8.2 | 3.1 | 0 | 13.4 | 4.1 | 93 |
| Beta-COM | bi. | 24 | 100 | 44.9 | 45.8 | 19.3 | 4.6 | 0 | 30.3 | 13.6 | 89 |
| ZSM-5-CON | bi. | 24 | 100 | 42.7 | 45.9 | 16.2 | 4.3 | 0 | 33.6 | 14.4 | 91 |
| ZSM-5-COM | bi | 24 | 100 | 59.4 | 40.5 | 21.4 | 9.7 | 0.3 | 28.1 | 16.7 | 81 |

[a] Phase: aqueous phase and biphasic phase represent by aq and bi., respectively, [b] Glucose, [c] Fructose, [d] Lactose, [e] Formic acid, [f] Levulinic acid, [g] Hydroxymethylfurfur al, [h] Yield of HMF (%), [i] Mass balance (%).

To compare the catalytic performances in the glucose conversion to HMF over the different zeolite frameworks and the commercial Beta, the glucose conversion is significantly lower when using the commercial Beta (Beta-COM), the conventional ZSM-5 (ZSM-5-CON), and the commercial ZSM-5 (ZSM-5-COM, as the characterization of this sample as shown in Figure S4 and Table S3) (44.9%, 42.7% and 59.4% for Beta-COM, ZSM-5-CON and ZSM-5-COM, respectively, and the HMF yield is 13.6%, 14.4%, and 16.7% for Beta-COM, ZSM-5-CON, and ZSM-5-COM, respectively). The decrease in the catalytic activity confirms the two effects: (i) the increase in the catalytic performance when using the bare Beta nanocrystals with respect to the commercial one, (ii) the incorporation of Sn acting as the additional Lewis active site. These finding open up the perspectives to design the hierarchical Beta nanocrystals with the incorporation of Sn in the framework, which can greatly improve the catalytic conversion of glucose to HMF in a biphasic system.

## 3. Experimental Section

### 3.1. Chemicals and Materials

$\alpha,\alpha'$-dichloro-p-xylene (98%, TCI), N,N,N′,N′-tetramethyl-1,6-hexanediamine (98%, TCI), sodium silicate ($Na_2Si_3O_7$: 26.5 wt% $SiO_2$, and 10.6 wt% $Na_2O$, Merck), aluminum sulphate ($Al_2(SO_4)_3 \cdot 18H_2O$, Univar, Ajax Finechem), sulfuric acid ($H_2SO_4$: 96%, RCI Labscan), tin(IV) chloride pentahydrate ($SnCl_4 5H_2O$, Sigma-Aldrich,), and sodium hydroxide (NaOH: 98%, Carlo Erba) were used as starting materials for the synthesis of Sn-incorporated Beta nanocrystals. Glucose (>97%, TCI), hydrochloric

acid (HCl: 37%, Merck), and dioxane ($C_4H_8O_2$, Merck) were used for the catalytic activity testing without any further purification. ZSM-5-CON zeolites were prepared by a one-pot hydrothermal process following the procedure reported in the literature [38], the commercial Beta zeolite was supplied from Zeolyst International (($NH_4$)BEA, Si/Al 12.5, CP814P) and the commercial ZSM-5 zeolite was supplied from Zeolyst International (($NH_4$)MFI, Si/Al 15, 3024E).

### 3.2. In Situ Synthesis of Sn-Beta Zeolite Nanocrystals

Firstly, the cyclic diquaternary ammonium (CDM) as a structure-directing agent (SDA) was synthesized according to the literature methods with some modification [37]. Briefly, N,N,N',N'-tetramethyl-1,6-hexanediamine was mixed with an equimolar amount of $\alpha,\alpha'$-dichloro-p-xylene in acetonitrile. The mixture was synthesized under 60 °C for 3 h in a round-bottom flask, controlling the temperature by oil bath. Subsequently, the solid powder was collected by filtering and washing with acetonitrile followed by diethyl ether. Finally, the solid CDM was dried in an oven at 100 °C for 12 h.

Secondly, the Sn-incorporated Beta zeolite nanocrystals were synthesized by using a one-pot hydrothermal process, in which sodium silicate and aluminum sulphate were used as silica and alumina sources, respectively. Meanwhile, the CDM was used as the structure-directing agent (SDA) to control the structure of Beta zeolite, and $SnCl_4$ was used as an Sn precursor. The molar composition was $xSnCl_4$: $30Na_2O$:$2.5Al_2O_3$:$100SiO_2$:$10CDM$:$15H_2SO_4$:$6000H_2O$ [46]. In a typical procedure, the Sn-Beta samples were prepared from two solutions. The first solution was 1 g of sodium silicate in a sodium hydroxide solution. To prepare the second solution, 0.13 g of aluminum sulphate in $H_2SO_4$ solution was obtained. Subsequently, the second solution was added dropwise to the first solution under vigorous stirring. Before stirring at room temperature for 1 h, the CDM and $SnCl_4$ were added into the mixed solution. Afterwards, the resultant mixture was transferred to a Teflon lined stainless steel autoclave for hydrothermal treatment at 170 °C for 24 h. The obtained product was filtered and washed with deionized water, dried at 100 °C overnight, and calcined at 550 °C for 6 h. The solid products were transformed to protonated form by ion exchanged with 1 M ammonium chloride solution at 80 °C for 2 h, repeated for three times. Then, the products were washed with deionized water, dried at 100 °C overnight, and calcined at 550 °C for 6 h. The samples are denoted as x%Sn-Beta where x corresponds to the Sn loading (0, 0.4 and 0.7 wt%).

### 3.3. Characterization of Catalysts

The powder X-ray diffraction (XRD) patterns were used to investigate the crystalline structure of catalysts by using a Bruker D8 ADVANCE instrument (Billerica, Massachusetts,. United States) with 0.02° step sizes and a scan rate of 1° min$^{-1}$ in the 2θ range of 5–60°. Wavelength-dispersive X-ray fluorescence spectrometry (WDXRF) was used to analyze the elemental composition of the catalysts and was performed using a Bruker S8 Tiger ECO instrument (Billerica, Massachusetts,. United States). Scanning electron microscopy (SEM) and transmission electron microscopy (TEM) were used to observe the surface morphology and topology of catalysts by performing on a JEOL JSM-7610F microscope (Tokyo, Japan) with an acceleration of 1 kV and a JEOL-JEM-ARM2000F microscope (Tokyo, Japan) operating at 200 kV, respectively. Energy dispersive spectroscopy (EDS) was used to measure the elemental distribution of catalysts by using a JEOL JSM-7610F microscope (Tokyo, Japan) operating at 15 kV. The textural properties of catalysts were determined via an $N_2$ adsorption/desorption technique at −196 °C, performed on a BELL-MAX analyzer (Tokyo, Japan). Prior to measurement, the catalysts were carefully degassed at 300 °C for 24 h with the temperature ramp rate of 20 °C min$^{-1}$ under the vacuum system ($10^{-1}$–$10^{-2}$ kPa). The specific surface area ($S_{BET}$) was calculated by using the Brunauer-Emmett-Teller (BET) theory. The total pore volume ($V_{total}$) was calculated at $P/P_0 = 0.99$. The t-plot method was used to estimate the micropore volume ($V_{micro}$), and the external volume ($V_{ext}$). The profiles of temperature-programmed desorption of ammonia ($NH_3$-TPD) using a BELL-CAT II analyzer (Tokyo, Japan) were performed to observe the acid properties (acid density and acid strength) of catalysts. The $NH_3$-TPD profiles were monitored in the temperature range of 100 to 700 °C with

a heating rate of 10 °C min$^{-1}$ in the flow of He. The X-ray photoelectron spectroscopy (XPS) depth profiles were recorded using a JEOL JPS-9010 (Tokyo, Japan) equipped with nonmonochromatic Mg K X-rays (1486.6 eV). An argon ion gun was used to etch the samples with the etching rate of 0.5 nm s$^{-1}$, and the XPS spectra were obtained at approximately 20 nm depth intervals. FTIR spectra of pyridine adsorption were recorded in the range of 4000–600 cm$^{-1}$ performed on a Bruker Invenio R (Ettlingen, Germany) instrument equipped with an MCT detector. The spectra were gained at a resolution of 4 cm$^{-1}$ with 64 scans. Before measuring spectra, the zeolite sample was pretreated by 10 v/v% of N$_2$ at 500 °C for 1 h with the temperature ramp rate of 10 °C min$^{-1}$. Subsequently, pyridine was introduced into the chamber at its vapor pressure at 40 °C for 1 h. After removing the physically adsorbed pyridine under vacuum for 1 h, the spectra were recorded at 150 °C [47].

### 3.4. Catalytic Testing

The synthesized catalysts were tested on glucose conversion under an aqueous and biphasic system (water/dioxane) with a small amount of HCl (pH = 1) performed in a high-pressure batch reactor. The reaction phase consists of a glucose (0.54 g, 3.6 wt%), catalyst (0.15 g), and deionized water (15 mL). In the case of the biphasic system, the solvent is the mixture of 13 mL of dioxane, and 2 mL of deionized water (deionized water phase was perfectly dissolved in organic phase). The experiment was carried out at different temperatures (100, 120, and 140 °C), autogenous pressures, and different time durations (4, 24, and 48 h). After the reaction, the liquid phases were separated from the solid phases (catalyst and any solid products formed) using a 0.45 μm membrane filter. High performance liquid chromatography (HPLC) with a Refractive index detector (RID) detector was used to analyze liquid products. Glucose and other products were monitored with an SH1011 sugar Shodex column, using MilliQ water (0.5 mM of H$_2$SO$_4$) as the mobile phase at a flow rate of 0.5 mL/min and a column temperature of 60 °C. The glucose conversion, selectivity, and yield of products are defined as follows:

$$Glucose\ conversion\ (\%) = \frac{mole\ of\ converted\ glucose}{mole\ of\ starting\ glucose} \times 100$$

$$Selectivity\ of\ products\ (\%) = \frac{mole\ of\ product}{total\ mole\ of\ products} \times 100$$

$$Yield\ of\ products\ (\%) = \frac{conversion \times selectivity}{100}$$

The mass balance of all experiments was calculated in the range of 61–93%.

To analyze the product in the organic phase, gas chromatography (GC) (Agilent, GC system 7890 B) (Agilent Technologies, Palo Alto, CA, USA) equipped with mass spectrometer (Agilent, system 5977A MSD) (Agilent Technologies, Palo Alto, CA, USA) with an HP-5MS capillary column (30 m, 0.32 mm i.d., stationary phase thickness of 0.25 μm) was performed to analyze the products of the organic phase from the reaction.

The Sn-Beta nanocrystals exhibit a bifunctional catalytic capability for the conversion of glucose to HMF. The incorporated Sn acting as Lewis acid sites can catalyze the isomerization of glucose to fructose. Subsequently, the Brønsted acid function can convert fructose to HMF via dehydration, as shown in Scheme 1.

**Scheme 1.** The conversion of glucose to HMF through the glucose isomerization and subsequent dehydration on bifunctional catalyst.

## 4. Conclusions

In the present study, the Sn-incorporated Beta nanocrystals have been successfully synthesized via an in-situ hydrothermal process. The Sn species in the Beta zeolite nanocrystals is an isomorphous species, which can be incorporated inside the zeolite framework. In addition, the synthesized Sn-Beta nanocrystals have been applied as bifunctional catalysts for converting glucose to HMF. These synthesized materials composed of Lewis and Brønsted acid sites generated by the incorporated Sn and Brønsted acid sites, respectively, providing the glucose isomerization to fructose on the Sn active sites and the dehydration of fructose to HMF on Brønsted acid sites. Furthermore, the effects of various parameters including Sn content, zeolite type, reaction time, reaction temperature, and solvent on the glucose conversion to HMF were investigated in the detail. It was found that using the 0.4 wt% Sn-Beta zeolite nanocrystals in a dioxane/$H_2O$ system at 120 °C for 24 h exhibited the greatest performance for producing HMF, with a yield of 42.0%. These findings open up the perspectives for the development of Sn incorporated in the Beta zeolite nanocrystals via an in-situ hydrothermal synthesis for the biomass conversion to high value-added chemical products.

**Supplementary Materials:** The following are available online at http://www.mdpi.com/2073-4344/10/11/1249/s1, Figure S1: Particle size distribution of: (a) the synthesized bare Beta, (b) the in-situ synthesized Sn incorporated Beta (0.4 wt% Sn-Beta), (c) the commercial Beta (Beta-COM), and (d) the conventional ZSM-5 (ZSM-5-CON), Figure S2: $N_2$ adsorption/desorption isotherms of (a) the synthesized bare Beta, and (b) the in-situ synthesized Sn incorporated Beta (0.4 wt% Sn-Beta), Figure S3: $NH_3$-TPD profiles of (a) the synthesized bare Beta, (b) the in-situ synthesized Sn incorporated Beta (0.4 wt% Sn-Beta), (c) the commercial Beta (Beta-COM), and (d) the conventional ZSM-5 (ZSM-5-CON), Figure S4: (A) XRD pattern (B) SEM image, (C) Particle size distribution and (D) $NH_3$-TPD profile of the commercial ZSM-5 (ZSM-5-COM) zeolite. Table S1: Chemical compositions analyzed by XRF of the synthesized bare Beta, the in-situ synthesized Sn incorporated Beta (0.4 wt% Sn-Beta), the commercial Beta (Beta-COM), and the conventional ZSM-5 (ZSM-5-CON), Table S2: Brønsted/Lewis acid site ratio was calculated by integrated area of main peaks, Table S3: Acid sites density of all samples determined via the ammonia temperature-programmed desorption ($NH_3$-TPD).

**Author Contributions:** Conceptualization, C.W.; Methodology, K.S., A.T., P.D., S.S., P.P.; Formal Analysis, K.S., A.T.; Investigation, S.N.; Writing—Original Draft Preparation, K.S., A.T.; Writing—Review and Editing, A.T., C.W. All authors have read and agreed to the published version of the manuscript.

**Funding:** This work was financially supported by the Vidyasirimedhi Institute of Science and Technology, and the National Research Council of Thailand (NRCT: Mid-Career Research Grant 2020). This research also received financial support from TSRI.

**Conflicts of Interest:** The authors declare no conflict of interest.

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
