# Peer review of "In Situ Synthesis of Sn-Beta Zeolite Nanocrystals for Glucose to Hydroxymethylfurfural (HMF)"

_catalysts, doi:10.3390/catal10111249_

Round 1

Reviewer 1 Report

The study describes the effect of the substitution of Si by Sn in BEA zeolite nanocrystals for the conversion of glucose to HMF. The influence of Sn quantity, zeolite structure, reaction time, reaction temperature and solvent is investigated. The results and interpretation are good for most of the parameters tested, but some points remain unclear. This work is within the scope of Catalysts and can probably be published provided that the following points are taken into account :

-Line 24 add a space between “nanocrystals” and “in”

-Line 36 correct compose by composed

-Line 65: correct “converts” by “convert”

-Line 67 : it is probably Sn that acts as the Lewis acid site and not the whole zeolite. Therefore, please add "in" after “incorporated” in the sentence “ … and Sn incorporated in Beta zeolite can act as Lewis acid sites.”

-Line 93: correct “has been” by “have been “ in “The … nanocrystals …. have been successively synthesized …”

-Line 99, suppress the dot after (Beta-COM) and before indicating

-Line 104-110: comparison is made between nano-sized BEA crystals (synthesized bare BEA, Sn BEA and commercial BEA, 21-38 nm) and a ZSM-5 of microscopic size 100 times larger (2500 nm).  If the objective was to compare the effect of the structure, it would be more appropriate to compare with ZSM-5 nanocrystals. The comparison BEA/ZSM-5 is therefore not convincing.

-Line 136-138: It is reported that the Si/Al ratios for the bare-BEA and Sn-BEA are 9 and 9.4, respectively, and it is claimed that Si sites are replaced by Sn in the Sn-BEA, leading to a higher Si/Al ratio in Sn-BEA than in the bare BEA. This is not clear because if Si is replaced by Sn at constant Al, Si decreases and Si/Al decreases and then it is not consistent with the sentence.

- Line 140 and table S1 : what is the relation between SiO2, Al2O3 and Si/Al values in table S1?  The values shown in the table S1 seems to be incorrect for ZSM-5. Indeed, by considering the formula for ZSM-5 which is  Mn/zz+ (AlO2)n(SiO2)96-n)yH2O,  I assume that 96-n corresponds to 89.4 in Table S1 which corresponds to SiO2 and therefore 6.62 should correspond to n which is AlO2 and not Al2O3 contrary to what is indicated in Table S1. The corresponding Si/Al ratio should be 13.5 and not 11.5 as it is written in table S1 and line 140. Please check yours values for ZSM-5 but also for BEA.

-Line 161: indicate the temperature instead of xx

-Line 163: correct slightly by slight

-Line 212: correct thea by the

-Line 217: correct respective by respectively

-Line 234 : suppress “in” before “which”

-Line 335: correct “compose of” by “composed of”

-Line 337 correct “on at” by “on”

-Before conclusion section, a diagram would be appreciated to describe the reaction mechanism in order to explain what is happening at the acidic and Bronsted acid sites.

Author Response

First of all, we are very appreciated to all comments from the reviewers.  The manuscript has been carefully revised following reviewers’ comments and the revised parts are highlighted in the manuscript, Please see the attachment.

We thank the referees for their very constructive comments and kind consideration,

Best regards,

Anawat Thivasasith

Reviewer 2 Report

The paper “In Situ Synthesis of Sn-Beta Zeolite Nanocrystal for Glucose to Hydroxymethylfurfural (HMF)” is a study on Sn substituted Beta nanocrystals synthesized by in-situ by hydrothermal process with the aid of cyclic di-quaternary ammonium as structure directing agent. This catalyst is a bifunctional catalyst for the conversion of glucose to HMF acting as Lewis acid sites and as Brønsted acid.

Authors characterized the samples by XRD, XPS, BET, NH3-TPD, pyridine adsorption by FTIR, SEM and TEM end studied the effects of Sn amount, zeolite type, reaction time, reaction temperature and solvent on the catalytic performances, finding good results using a mixure dioxane/H2O as solvent.

The paper is not very extensive but coherent with the purposes and complete. The characterization is appropriate to the materials and to the catalytic process under study. The results clearly presented. The introduction is comprehensive. The conclusions are justified by the results.

The paper can be accepted with minor revisions.

Minor revisions:

I suggest shortening the name in the text of the prepared sample “in-situ synthesized Sn incorporated Beta nanocrystals” repeated many times in the text and too long to read, using only “ 0.4 wt%Sn-Beta” or “0.7 wt%Sn-Beta”; “the synthesized bare Beta” shortened in “bare Beta”.

Line 161: add temperatures

Author Response

(The authors gave the same response as above.)
